# Static and Free Vibration Analyses of Functionally Graded Plane Structures

J.S.D. Gaspar [1,*], M.A.R. Loja [1,2,*] and J.I. Barbosa [1,2,*]

1   CIMOSM—Centro de Investigação em Modelação e Otimização de Sistemas Multifuncionais, ISEL, IPL—Instituto Politécnico de Lisboa, Av. Conselheiro Emídio Navarro 1, 1959-007 Lisboa, Portugal

2   IDMEC, Instituto Superior Técnico—Universidade de Lisboa, Av. Rovisco Pais, 1, 1049-001 Lisboa, Portugal

*   Correspondence: a44997@alunos.isel.pt (J.S.D.G.); amelia.loja@isel.pt (M.A.R.L.); joaquim.barbosa@tecnico.ulisboa.pt (J.I.B.)

**Abstract:** In recent years, the use of functionally graded materials has been the focus of several studies due to their intrinsic ability to be tailored according to the requirements of structures while minimising abrupt stress transitions commonly found in laminated composites. In most studies, the materials' mixture gradient is established through a structural component, i.e., thickness, which is known to visibly enhance structural behaviour. However, depending on the type of structure, it is important to exploit the possibility of building a structure using other gradient directions. The innovative characteristic of this work, which aims to study plane truss and frame-type structures made of functionally graded materials, lies in the specificity that the materials' mixture gradient occurs as a function of a geometric structure feature, i.e., for example, the structure height, rather than the more usual approach, as a component dependence, i.e., through a member thickness or even along its length. The performance of the present model is illustrated through a set of case studies, and where possible, the results achieved are compared with more traditional solutions.

**Keywords:** functionally graded materials; structure-dependent material gradient; plane trusses; plane frames; static and free vibration analyses





## 1. Introduction

In engineering applications, conflicting property requirements necessitate combining materials, which can be in the molten state (or alloying), limited by thermodynamic equilibrium limits, and in the solid state, resulting in composite materials. Composite materials can be divided into laminates and functional gradients, with the latter being characterised by the minimisation of problems that occur in laminates, namely sharp interfaces between layers in which abrupt stress transitions occur and failures start [1,2].

Functionally graded materials are composite materials with properties that result from a combination of distinct materials in which the mixture proportions vary along one or more spatial coordinates. The improved properties of FGMs that have been achieved by using this tailoring flexibility in parallel with technological advances in manufacturing such as additive manufacturing, powder metallurgy, etc., (Kumar et al. [3]) justify their increasingly frequent use. FGMs have wide applicability in aircraft, spacecraft, and space structures, which space exploration depends on, as well as in car parts, electronic and medical equipment, thermal coatings for ceramic engines, the marine industry, gas turbines, and biomedicine, among other applications [4–8]. Although functionally graded materials still have high production costs, this type of material is a topic that has been significantly discussed by numerous authors in recent years, with many publications on the subject being published, making it a very current and important topic that should be explored further [1,9,10].

To investigate the advantages of using FGMs, several authors have developed models through which static and free vibration analyses have been carried out to investigate the

influence of the materials' characteristic parameters on their behaviour. In this sense, Chi and Chung [11] investigated elastic, rectangular, and simply supported FGM plates of medium thickness that were subjected to transverse loading. They considered constant Poisson's ratios and varying the elasticity modulus through the plate's thickness according to a power law, i.e., sigmoid, or exponential volume fraction function. The series solutions were obtained based on the classical plate theory and Fourier series expansion. The closed-form solutions illustrated by the Fourier series expression were given in this paper, and the closed-form and finite element solutions were compared and discussed in the second part of the study (Chi and Chung [12]).

Reddy et al. [13] published a review paper about the governing equations and analytical solutions of the classical and shear deformation theories of FGM beams. More specifically, the classical, first-order, and third-order shear deformation theories accounted for the through thickness variation in a dual-phase FGM and modified couple stress (i.e., strain gradient) and von Kármán nonlinearity. This study also included analytical solutions for the bending of the linear theories to illustrate the influence of material variation, boundary conditions, and loads. One of the major issues when manufacturing FGMs is the presence of porosities. Considering this, Jouneghani et al. [14] investigated the vibrational behaviour of doubly curved shells made of FGMs, including porosities. The first-order shear deformation theory was taken as the theoretical framework, considering that the FGMs' mechanical properties would vary through the thickness direction according to two power laws of volume fraction distribution. The strain components were established in an orthogonal curvilinear coordinate system, and the governing equations were derived according to Hamilton's principle, and then Navier's solution method was used; the numerical results concerning three different types of shell structures were presented. Pham et al. [15] proposed the notion of combining the edge-based smoothed finite element method and mixed interpolation of tensorial components' triangular element to carry out a dynamic analysis of sandwich plates with a functionally graded porous core subjected to blast load. The configuration of the sandwich plates considered a homogeneous metal bottom layer, an all-ceramic top layer, and a FGP core layer with uneven porosity distribution. The proposed element aimed to enhance the accuracy and convergence of the MITC3 element as well as the traditional triangular element. The authors performed some studies to confirm the expected performance of the proposed method and other numerical studies to characterize the vibration behaviour of the sandwich plates.

Chakraborty et al. [16] verified that the presence of FGM layers in beams produced significantly different behaviours when compared with other beams constituted by only one of the constituent materials due to the combination of their properties. Koutoati et al. [17] studied the stresses and strains in the interlayer interfaces of a sandwich component, and they have found that the use of FGM allows for a reduction in the presence of residual stresses, thus minimizing the occurrence of delamination and microcracking at the interfaces. In a study by Soliman et al. [18], isotropic and orthotropic beams with functional gradient were studied, considering the variation in properties through the thickness. The results showed that the FGM beams with different power law exponents have greater resistance to distributed loads and concentrated loads when compared to isotropic and orthotropic beams for different boundary conditions and different lengths. Wang et al. [19] investigated the dynamic behaviour of a pinned–pinned spinning exponentially functionally graded shaft with unbalanced loads. For this, they used the Rayleigh beam model, considering rotary inertia and gyroscopic effects. The model validity was confirmed, and a numerical parametric study was carried out to assess the influence of the main parameters. The results achieved led to the conclusion that the vibration and instability of the spinning shaft strongly depended on the unbalanced load and material gradient.

In a study by Zohra et al. [20], the critical buckling temperatures and natural frequencies of beams with a thickness functional gradient were investigated, and it was observed that the effects of shear deformation on frequencies tend to be more significant when the beams become shorter (thicker). These effects were more evident for higher

mode frequencies. In a study by Singh et al. [21], it was stated that most FGMs presented material properties variation through the thickness direction and also in the axial direction, although only to a minor extent. In addition, the authors noted that the selection of a suitable material for the intended application was an immediate and direct challenge for the future development of technology in this field of research. In a study by Cao et al. [22], there were studied beams with uniform and non-uniform cross-sections with an axially functionally graded material for different boundary conditions. In Maalawi's study [23], an optimisation model for improving the performance of different types of structural composites was presented, and the FGM concept was also taken into account. Several scenarios were considered to model the spatial variation in these materials' properties. The proposed optimisation strategies included maximising natural frequencies in thin composite beams, optimising transmission shafts against torsional buckling and rotational instability, and maximising the critical flight speed of subsonic aircraft wings. Wu et al. [24] developed a mixed finite element for the nonlinear free vibration analysis of FGM beams based on the Timoshenko beam theory under combinations of simply supported, free, and clamped edge conditions. Additionally, the finite element considered the von Kármán geometrical nonlinearity. The material properties of the beam varied through the thickness according to power-law volume fraction distributions, and the effective material properties were estimated via the rule of mixtures. A multilayer perceptron back propagation neural network was also developed to predict the nonlinear free vibration behaviour of the FGM beam.

In Katili et al.'s study [25], it was found that, in functionally graded beams with varying properties through the thickness, the evolution of the modulus of elasticity has the greatest influence on the beam's stress and displacement distributions. Thus, choosing the power law exponent adequately allowed the material properties to be modelled to comply with the requirements regarding the minimisation of stresses and displacements in beams. Alshorbagy et al. [26] analysed the free vibration characteristics and dynamic behaviour of beams with variation in properties along the thickness and axial directions using the finite element method. It was verified that the variation in the material distribution along the axial direction influenced the variation in the beam stiffness along its length, hence affecting the frequencies and the respective vibration modes. However, the vibration modes were not affected by the variation in properties in thickness. This study also found that the natural frequencies increased with increasing power law exponent when the ratio between the modulus of elasticity was less than 1 and decreased otherwise.

Murin et al. [27] presented a homogenised FGM beam finite element for modal analysis, considering a double symmetric cross-section. The element considered the shear deformation effect and the effect of longitudinally varying inertia and rotary inertia as well as a longitudinally varying Winkler elastic foundation and the influence of internal axial forces. The material effective quantities assumed a continuous variation and were determined using extended mixture rules and the multilayer method. The authors performed numerical simulations via conducting modal analyses of single FGM beams and spatial beam structures. Nguyen et al. [28] considered the first-order shear deformation theory to carry out static and free vibration analyses of axially loaded rectangular FGM beams. Improved transverse shear stiffness was derived from the in-plane stress and equilibrium equation; thus, the associated shear correction factor was obtained analytically. The authors presented analytical solutions for simply supported FGM beams, and the results were compared with existing solutions. The influence of the power-law exponent, the dissimilarity of the constituent materials, and the Poisson's ratio was investigated through different mechanical responses. A finite element model based on the first-order shear deformation theory for free vibration and the buckling of functionally graded beams was proposed by Kahya and Turan [29]. The material properties varied continuously through the beam thickness according to the power-law expression, and the governing equations were derived via Lagrange's equations. Numerical parametric studies for the natural frequencies and buckling loads were performed for different boundary conditions, power-law exponents, and span/depth ratios.

Banerjee and Ananthapuvirajah [30] studied the free vibration behaviour of FGM beams (FGBs) and frameworks containing FGBs by using the dynamic stiffness method. In their study, it was considered that the material properties of the FGBs would vary continuously through the thickness according to a power law function. The numerical results were validated against alternative published results, and as mentioned by the authors, in the absence of published results for frameworks containing FGBs, consistency checks on the reliability of results were performed.

From our review of the literature, which has been briefly summarized in the above paragraphs, it was possible to understand that there are many studies about the behaviour of beams, plates, and even shells in FGMs; however, it was not possible to find published investigations on the behaviour of plane truss and/or frame-type structures, where the material gradient is associated with a structure dimensional characteristic, such as its height or its width.

One considers that this structure-dependent material gradient approach can be viewed as an additional design variable when dealing with such types of structures, so this was the main motivation to adopt a material gradient evolution that depends on the structure's vertical coordinate.

The first section of the present study considers some aspects related to the development of the proposed finite element model. This is followed by a section where the model is verified for some isotropic homogeneous and FGM beams. The results achieved are then compared against alternative solutions. Afterwards, several plane truss and frame structures, are analysed and discussed.

To the authors' knowledge, aside from the set of case studies made available for possible reproduction, there are no published works that consider this paper's proposed approach, and this constitutes the main and innovative contribution of the present work.

## 2. Materials and Methods

### 2.1. Functionally Graded Materials

The prediction of the macroscopic properties of an FGM can be modelled according to different homogenisation models [31], perhaps the most used of which is the Rule of Mixtures. According to this rule, if one has a non-porous FGM composed by material 1 and material 2, a generic property $P_{FGM}$, can be predicted by Equation (1), where $V_f$ corresponds to the volume fraction of material 1 (in the present work, a metallic or ceramic material). The volume fraction can be defined as a function of different space coordinates. Figure 1 shows the evolution of the volume fraction for different power law exponents.

$$P_{FGM}(x,y,z) = P_1 V_f + P_2 \left(1 - V_f\right) \tag{1}$$

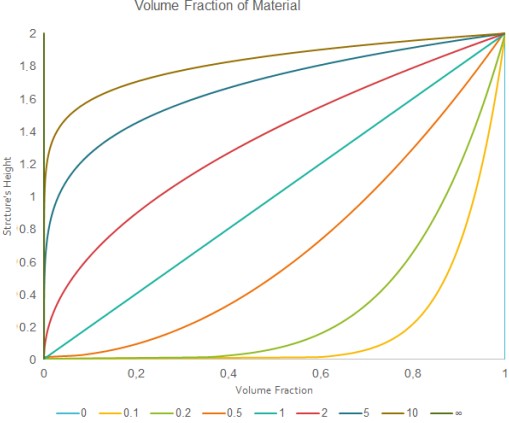

**Figure 1.** Volume fraction of material 1 as a function of a structure's height.

In most published works that focus on beam structures, the volume fraction is expressed in terms of the thickness or the length of a beam. For illustrative purposes, Figure 2a and Equation (2) represent a through thickness mixture variation (direction *z*) that considers the typical power law.

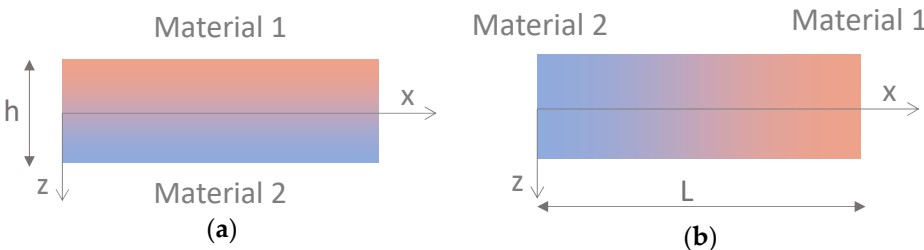

**Figure 2.** Structure with variation in its properties (**a**) through the thickness and (**b**) along the length.

Figure 2b and Equation (3) illustrate a material's mixture along the *x* direction that is associated with the length of a beam and its volume fraction law [32].

$$V_f = \left( \frac{z}{h} + \frac{1}{2} \right)^p \tag{2}$$

$$V_f = \left( \frac{x}{L} \right)^p \tag{3}$$

As the focus of this work is the study of 2D structures with functional gradients relative to the vertical coordinate of a structure, a generic structure in which this variation in properties occurs is presented in Figure 3.

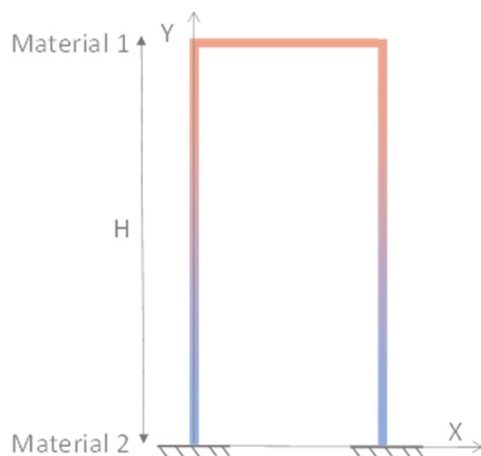

**Figure 3.** Structure with variation in its properties as a function of its height.

For this case, considering H as the maximum height of the structure, the volume fraction can be generically written as follows:

$$V_f = \left( \frac{Y}{H} \right)^p \tag{4}$$

### 2.2. Displacement Field and Equilibrium Equations

To carry out this study, we used the First-order Shear Deformation Theory (FSDT), and the displacement field for a beam under bending in the *xy* plane can be written as follows [32,33]:

$$\begin{Bmatrix} u(x,y,t) \\ v(x,t) \end{Bmatrix} = \begin{Bmatrix} u^0(x,t) \\ v_0(x,t) \end{Bmatrix} + y \begin{Bmatrix} \theta_z^0(x,t) \\ 0 \end{Bmatrix} \tag{5}$$

The beam schematic's representation and its relation to a local coordinate system, as well as its degrees of freedom, can be observed in Figure 4.

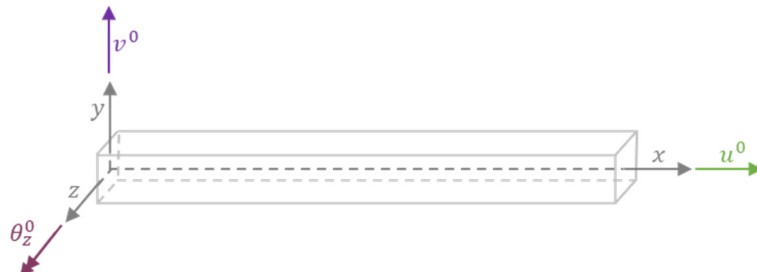

**Figure 4.** Beam representation and local coordinate system.

The degrees of freedom, $u^0$ and $v^0$, are the mid-plane displacements along the $x$ and $z$ directions, which correspond to the length and thickness directions of the beam. $\theta_z^0$ is the rotation of the beam's mid-plane around the direction $z$.

The corresponding constitutive relations are in a compact form and can be written as follows [32,33]:

$$\begin{Bmatrix} \sigma_x \\ \sigma_{xy} \end{Bmatrix} = \begin{bmatrix} Q_{11} & 0 \\ 0 & Q_{66} \end{bmatrix} \begin{Bmatrix} \varepsilon_x \\ \gamma_{xy} \end{Bmatrix} \tag{6}$$

where the generalized strain vectors, considering small deformations, are as follows:

$$\{\varepsilon_{xx}\} = \begin{bmatrix} \frac{\partial}{\partial x} & 0 & y\frac{\partial}{\partial x} \end{bmatrix}\{q\}\{\gamma_{xy}\} = \begin{bmatrix} 0 & \frac{\partial}{\partial x} & 1 \end{bmatrix}\{q\} \tag{7}$$

with the generalized displacements vector $\{q\}$ consisting of the degrees of freedom $u_0$, $v_0$, and $\theta_z$, and the membrane-bending and shear elastic stiffness coefficients matrices are given as

$$Q_{mb} = [Q_{11}] = [E_{FGM}] \tag{8}$$

$$Q_s = [Q_{66}] = \left[ \frac{E_{FGM}}{2(1 + v_{FGM})} ks \right] \tag{9}$$

Considering the aim of the present work—focusing on the static and free vibration analyses of FGM 2D truss and frame-type structures—we considered the Hamilton principle to derive the equilibrium equations. Generically, this principle's formulation is given as shown in Equation (10) [34]:

$$\delta \int_{t_1}^{t_2} [T - (W + U)]dt = 0 \tag{10}$$

where, for a generic finite element ($e$), the respective values of the kinetic energy $T$, the elastic strain energy $U$, and the work performed by the external applied surface forces $W$ are given as follows:

$$T = \tfrac{1}{2} \int_{Ve} \left( \rho_{FGM} \left( \dot{u}^2 + \dot{v}^2 \right) \right) dVe$$
$$U = \tfrac{1}{2} \int_{Ve} \left( \{\varepsilon_{xx}\}^t [Q_{mb}]\{\varepsilon_{xx}\} + \{\gamma_{xy}\}^t [Q_s]\{\gamma_{xy}\} + \right) dVe \tag{11}$$
$$W = \int_{Se} \left( \{q\}^t \{f_s\} \right) dSe$$

After some mathematical manipulation, one can yield the corresponding equilibrium equation at the element level, which, in a compact written format, can be written as follows:

$$[M^e]\{\ddot{q}^e\} + [K^e]\{q^e\} = \{F^e\} \tag{12}$$

For the linear static analysis, the contribution of the kinetic energy is neglected; thus, the equation is simplified, leading to the following:

$$[K^e]\{q^e\} = \{F^e\} \tag{13}$$

Otherwise, to perform free vibration analysis, only the load vector will be null. Assuming free harmonic vibrations, the equilibrium equation will become Equation (14) [31,34,35]:

$$\left([K^e] - \omega^2[M^e]\right)\{q^e\} = \{0\} \tag{14}$$

where $K^e$, $M^e$, and $\{F^e\}$ represent the element stiffness, mass matrices, and element forces vector, respectively; $q^e$ and $\ddot{q}^e$ represent the element nodal degrees of freedom and corresponding accelerations; and $\omega$ denotes the j-th natural frequency.

To implement these equilibrium equations in the discretized model context, we used Lagrange quadratic beam elements with three degrees of freedom at a generic k-th node—the axial ($u_k$) and transverse ($v_k$) displacement and the rotation on the plane $xy$ ($\theta_{zk}$), $k = 1,3$—associated with the midplane surface of the beam [34]. Figure 5 represents the one-dimensional quadratic element on its own reference in natural coordinates.

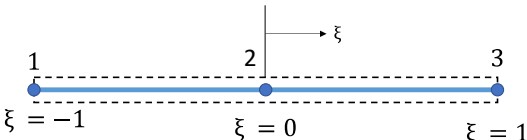

**Figure 5.** One-dimensional quadratic element on its natural coordinate system.

Considering that a generic degree of freedom is approximated as a linear combination of the corresponding nodal degrees of freedom and the shape functions, the previous element stiffness $K^e$ and mass $M^e$ matrices can be represented in a compact form as follows [34,36]:

$$K^e = \int_{Ve} B_{mb}^T \, Q_{mb} \, B_{mb} dVe + \int_{Ve} B_s^T \, Q_s \, B_s dVe \tag{15}$$

$$M^e = \int_{Ve} L^T \, \rho_{FGM} L dVe \tag{16}$$

where $B_{mb}$ and $B_s$—the matrices that pair the deformations with the displacements—and $L$—the matrix of the Lagrange quadratic shape functions $N_i$ that are associated with the displacement field description—are, for the i-th node, constituted as follows:

$$
\begin{aligned}
B_{mb} &= \begin{bmatrix} \frac{dN}{dx} & 0 & y \cdot \frac{dN}{dx} \end{bmatrix}_i \\
B_s &= \begin{bmatrix} 0 & \frac{dN}{dx} & N \end{bmatrix}_i \\
L &= \begin{bmatrix} N & 0 & y \cdot N \\ 0 & N & 0 \end{bmatrix}_i
\end{aligned} \tag{17}
$$

Considering that the volume fraction expression depends on the structure height, it must be first rewritten within the domain of each element. Hence, the volume fraction expression can be rewritten as in Equation (18), where, in the structure referential, $Y_{k,1}$ and $Y_{k,3}$ correspond to the height coordinates of the initial and final nodes of the kth element, respectively, and $\xi$ is the natural coordinate of the element local referential.

$$Y = (\xi + 1)\frac{Y_{k,3} - Y_{k,1}}{2} + Y_{k,1} \tag{18}$$

After the calculation of all elements' matrices and vectors, some may need to be transformed from their local coordinate system to the structure one, the global coordinate system, and when this happens, one should consider the usual transformation procedure [34]:

$$\begin{aligned} \boldsymbol{K}_{eg} &= \boldsymbol{T}^T \boldsymbol{K}_e \boldsymbol{T} \\ \boldsymbol{M}_{eg} &= \boldsymbol{T}^T \boldsymbol{M}_e \boldsymbol{T} \end{aligned} \tag{19}$$

Here, $\boldsymbol{T}$ is, in this case, the transformation matrix from the element to the structure coordinate system. The equilibrium equations' systems for the whole discretized domain are then obtained by assembling the element matrices and vector contributions, after which the boundary conditions are imposed, and the analyses can be performed.

## 3. Results

In the following sub-sections, we present some verification studies and, subsequently, a set of case studies to assess the influence of parameters that characterize the FGM plane structures.

### 3.1. Verification Studies

For each verification case presented in this section, the deviations in the obtained results compared to the ones from the literature were determined through using the following generic expression:

$$Deviation\ (\%) = \frac{Calculated\ Value - Reference\ Value}{Reference\ Value} \times 100 \tag{20}$$

### 3.1.1. Static Analysis of a Simply Supported Functionally Graded Beam

In the first verification case, we considered the behaviour of a simply supported functionally graded beam with an aspect ratio of L/h = 20, where L represents beam length, and h represents section height (also studied by Nam et al. [37]). The beam is made of aluminium ($E_m = 70$ GPa, $\nu = 0.3$) and alumina ($E_c = 380$ GPa, $\nu = 0.3$), and the non-dimensional transverse displacement can be obtained using the following multiplier:

$$\overline{w} = \frac{100\ E_m h^3}{qL^4}\ w\left(\frac{L}{2}\right) \tag{21}$$

For this purpose, a square cross-section with a unit edge and a twenty-fold higher length was assumed. The beam is subjected to a uniformly distributed load (q) of intensity of 100 N/m. The beam properties vary through the thickness according to the power law, and the rule of mixtures, which allows for the expression of this variation profile, is written as follows:

$$P(z) = (P_c - P_m)\left(\frac{z}{h} + \frac{1}{2}\right)^p + P_m \tag{22}$$

From Table 1, good agreement between the results obtained by the present model and the ones in [37] can be observed for all of the power law exponents.

**Table 1.** Non-dimensional transverse maximum deflection.

| $p$ | Nam et al. [37] | Present | Deviation (%) |
|---|---|---|---|
| 0 | 2.8962 | 2.8963 | 0.0035 |
| 0.5 | 4.4648 | 4.4648 | 0.0000 |
| 1 | 5.8049 | 5.8049 | 0.0000 |
| 2 | 7.4397 | 7.4397 | 0.0000 |
| 5 | 8.8069 | 8.8068 | −0.0011 |
| 10 | 9.6767 | 9.6768 | 0.0010 |

3.1.2. Free Vibration Analysis of a Simply Supported Functionally Graded Beam

The second verification case consists of a free vibration analysis of a simply supported functionally graded beam, considering a volume fraction distribution that varies both through the thickness and the length of the beam. Alshorbagy et al. [26] tested different aspect ratio relations and boundary conditions, as well as different power law exponent values. For the present work, we compared the present model's results with those of [26] for an aspect ratio of L/h = 20.

For the case of the functional gradient beam with variation in thickness, the rule of mixtures represents the beam being purely metallic on the bottom surface (*L* subscript) and purely ceramic at the top surface (*U* subscript), as given by the following:

$$P(z) = (P_U - P_L)\left(\frac{z}{h} + \frac{1}{2}\right)^{ez} + P_L \tag{23}$$

For the case of the functional gradient along the length, the rule of mixtures represents the beam being purely ceramic at the left extremity and purely metallic at the right extremity, according to the following expression:

$$P(x) = (P_L - P_R)\left(1 - \frac{x}{L}\right)^{ex} + P_R \tag{24}$$

The results obtained, have been adimensionalised, through the following relations:

$$\overline{x} = \frac{x}{L} \tag{25}$$

$$E_{ratio} = \frac{E_U}{E_L} \tag{26}$$

$$\rho_{ratio} = \frac{\rho_U}{\rho_L} \tag{27}$$

$$\lambda^2 = \omega L^2 \sqrt{\frac{\rho_L}{E_L} \frac{A}{I}} \tag{28}$$

The first fundamental frequencies for the functionally graded beam with variational properties through the thickness are given for the different elasticity moduli ratios in Table 2.

**Table 2.** Non-dimensional fundamental frequencies for functionally graded beams with variational properties through the thickness.

| ez | | Eratio | | | | |
|---|---|---|---|---|---|---|
| | | 0.25 | 0.5 | 1 | 2 | 4 |
| 0 | Present | 2.2168 | 2.6362 | 3.135 | 3.7282 | 4.4336 |
| | Authors [26] | 2.2203 | 2.6404 | 3.14 | 3.7341 | 4.4406 |
| | Deviation (%) | 0.1576 | 0.1591 | 0.1592 | 0.1580 | 0.1576 |
| 0.1 | Present | 2.3707 | 2.7064 | 3.135 | 3.6715 | 4.3299 |
| | Authors 26] | 2.3746 | 2.7107 | 3.14 | 3.6773 | 4.3366 |
| | Deviation (%) | 0.1642 | 0.1586 | 0.1592 | 0.1577 | 0.1545 |
| 0.2 | Present | 2.4572 | 2.7531 | 3.135 | 3.6243 | 4.239 |
| | Authors [26] | 2.4614 | 2.7576 | 3.14 | 3.63 | 4.2455 |
| | Deviation (%) | 0.1706 | 0.1632 | 0.1592 | 0.1570 | 0.1531 |
| 0.5 | Present | 2.5936 | 2.8317 | 3.135 | 3.5241 | 4.0284 |
| | Authors [26] | 2.5979 | 2.8363 | 3.14 | 3.5296 | 4.0346 |
| | Deviation (%) | 0.1655 | 0.1622 | 0.1592 | 0.1558 | 0.1537 |
| 1 | Present | 2.6998 | 2.8901 | 3.135 | 3.4369 | 3.8181 |
| | Authors [26] | 2.7041 | 2.8946 | 3.14 | 3.4423 | 3.8241 |
| | Deviation (%) | 0.1590 | 0.1555 | 0.1592 | 0.1569 | 0.1569 |

**Table 2.** *Cont.*

| ez | | Eratio | | | | |
|---|---|---|---|---|---|---|
| | | **0.25** | **0.5** | **1** | **2** | **4** |
| | Present | 2.8016 | 2.9417 | 3.135 | 3.3712 | 3.6433 |
| 2 | Authors [26] | 2.8057 | 2.9461 | 3.14 | 3.3768 | 3.6496 |
| | Deviation (%) | 0.1461 | 0.1493 | 0.1592 | 0.1658 | 0.1726 |
| | Present | 2.9261 | 3.0066 | 3.135 | 3.3139 | 3.5257 |
| 5 | Authors [26] | 2.9302 | 3.011 | 3.14 | 3.3196 | 3.5326 |
| | Deviation (%) | 0.1399 | 0.1461 | 0.1592 | 0.1717 | 0.1953 |
| | Present | 3.0042 | 3.0517 | 3.135 | 3.267 | 3.4481 |
| 10 | Authors [26] | 3.0085 | 3.0563 | 3.14 | 3.2726 | 3.4549 |
| | Deviation (%) | 0.1429 | 0.1505 | 0.1592 | 0.1711 | 0.1968 |

The first fundamental frequencies for the functionally graded beams with length-based variational properties are presented in Table 3.

**Table 3.** Non-dimensional fundamental frequencies for functionally graded beams with variation in properties along their length.

| ex | | Eratio | | | | |
|---|---|---|---|---|---|---|
| | | **0.25** | **0.5** | **1** | **2** | **4** |
| | Present | 2.2168 | 2.6362 | 3.135 | 3.7282 | 4.4336 |
| 0 | Authors [26] | 2.2203 | 2.6404 | 3.14 | 3.7341 | 4.4406 |
| | Deviation (%) | 0.1576 | 0.1591 | 0.1592 | 0.1580 | 0.1576 |
| | Present | 2.325 | 2.6826 | 3.135 | 3.6928 | 4.3696 |
| 0.1 | Authors [26] | 2.3285 | 2.6868 | 3.14 | 3.6988 | 4.3768 |
| | Deviation (%) | 0.1503 | 0.1563 | 0.1592 | 0.1622 | 0.1645 |
| | Present | 2.4068 | 2.7216 | 3.135 | 3.6593 | 4.3072 |
| 0.2 | Authors [26] | 2.4106 | 2.7258 | 3.14 | 3.6653 | 4.3144 |
| | Deviation (%) | 0.1576 | 0.1541 | 0.1592 | 0.1637 | 0.1669 |
| | Present | 2.5777 | 2.8103 | 3.135 | 3.5698 | 4.1311 |
| 0.5 | Authors [26] | 2.5821 | 2.8148 | 3.14 | 3.5758 | 4.1387 |
| | Deviation (%) | 0.1704 | 0.1599 | 0.1592 | 0.1678 | 0.1836 |
| | Present | 2.748 | 2.9056 | 3.135 | 3.4554 | 3.8863 |
| 1 | Authors [26] | 2.7533 | 2.9104 | 3.14 | 3.4611 | 3.8937 |
| | Deviation (%) | 0.1925 | 0.1649 | 0.1592 | 0.1647 | 0.1901 |
| | Present | 2.9218 | 3.0069 | 3.135 | 3.3192 | 3.5734 |
| 2 | Authors [26] | 2.9278 | 3.0122 | 3.14 | 3.3244 | 3.5795 |
| | Deviation (%) | 0.2049 | 0.1760 | 0.1592 | 0.1564 | 0.1704 |
| | Present | 3.077 | 3.0996 | 3.135 | 3.1877 | 3.2623 |
| 5 | Authors [26] | 3.0834 | 3.1052 | 3.14 | 3.1923 | 3.2668 |
| | Deviation (%) | 0.2076 | 0.1803 | 0.1592 | 0.1441 | 0.1377 |
| | Present | 3.1206 | 3.1262 | 3.135 | 3.1486 | 3.1684 |
| 10 | Authors [26] | 3.1265 | 3.1316 | 3.14 | 3.1531 | 3.1726 |
| | Deviation (%) | 0.1887 | 0.1724 | 0.1592 | 0.1427 | 0.1324 |

From Tables 2 and 3, good agreement between the results obtained through the implemented model and the ones from [26] can be observed for each material distribution (both for thickness and length), modulus of elasticity ratio, and power law exponent.

### 3.1.3. Static and Free Vibration Analysis of an Isotropic Frame-Type Structure

The isotropic frame-type structure presented in Figure 6a is discretized into six beam elements in which each element has three nodes, as can be observed in Figure 6b.

For the static analysis of the structure, the elements were considered to be made of a metallic material with an elasticity modulus of 200 GPa and a square cross-section with a 0.02 m edge. The length L was set to 1 m, and the magnitudes of loads were set to $Px = 100$ N, $Py = 1000$ N, $M0 = 250$ N.m, and $q = 90$ N/m.

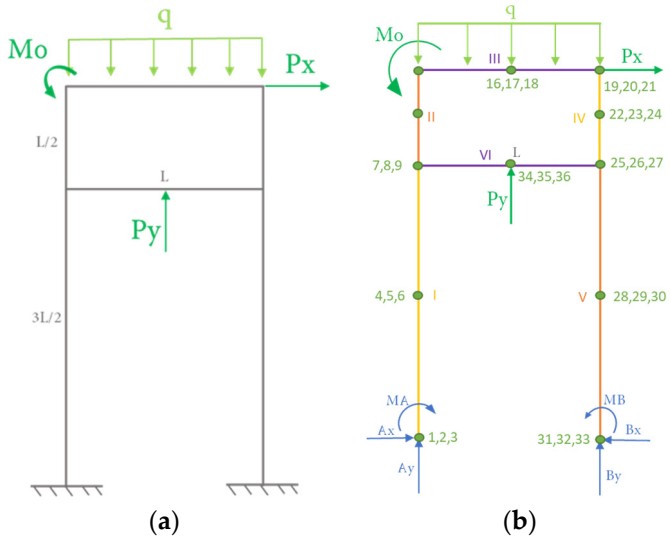

**Figure 6.** Schematic representation of (**a**) an isotropic frame-type structure and the loadings considered in the verification study and (**b**) the structure's minimum discretisation.

Different discretisations were considered by dividing the members (I, II, III, IV, V, VI) of the structure in such a way that the lengths of the elements were close to each other. The support reactions obtained are presented in Tables 4 and 5.

**Table 4.** Convergence study on the reactions in the clamped support at point A.

| Elements per Structure Members | | | | | | Reaction | | |
|---|---|---|---|---|---|---|---|---|
| I | II | III | IV | V | VI | Ax [N] | Ay [N] | MA [Nm] |
| 1 | 1 | 1 | 1 | 1 | 1 | 44.7968 | 73.5445 | 36.6813 |
| 2 | 2 | 2 | 2 | 2 | 2 | 44.7967 | 73.5432 | 36.6814 |
| 3 | 1 | 2 | 1 | 3 | 2 | 44.8049 | 73.5311 | 36.6878 |
| 6 | 2 | 4 | 2 | 6 | 4 | 44.8342 | 73.4699 | 36.7131 |
| 9 | 3 | 6 | 3 | 9 | 6 | 44.8350 | 73.4990 | 36.7094 |

**Table 5.** Convergence study on the reactions in the clamped support at point B.

| Elements per Structure Members | | | | | | Reaction | | |
|---|---|---|---|---|---|---|---|---|
| I | II | III | IV | V | VI | Bx [N] | By [N] | MB [Nm] |
| 1 | 1 | 1 | 1 | 1 | 1 | 55.2132 | 183.5445 | 41.8804 |
| 2 | 2 | 2 | 2 | 2 | 2 | 55.2131 | 183.5435 | 41.8805 |
| 3 | 1 | 2 | 1 | 3 | 2 | 55.2214 | 183.5313 | 41.8870 |
| 6 | 2 | 4 | 2 | 6 | 4 | 55.2505 | 183.4686 | 41.9121 |
| 9 | 3 | 6 | 3 | 9 | 6 | 55.2522 | 183.5039 | 41.9089 |

To verify the results obtained using the present model, we considered the analytical solution provided through Castigliano's second theorem and a numerical solution obtained by using an open-source finite element application named Ftool [38] (Table 6).

**Table 6.** Reactions in the boundary conditions of the frame-type structure obtained through using Castigliano's second theorem and the free-to-use Ftool application.

| Reaction | Castigliano's Second Theorem | Deviation (%) | Ftool [38] | Deviation (%) |
|---|---|---|---|---|
| Ax [N] | 45.1041 | −0.6813 | 44.7438 | 0.1185 |
| Ay [N] | 73.4626 | 0.1115 | 73.5544 | −0.0135 |
| MA [Nm] | 36.7870 | −0.2873 | 36.6529 | 0.0775 |

**Table 6.** *Cont.*

| Reaction | Castigliano's Second Theorem | Deviation (%) | Ftool [38] | Deviation (%) |
|---|---|---|---|---|
| Bx [N] | 54.8959 | 0.5780 | 55.2562 | −0.0778 |
| By [N] | 183.4626 | 0.0446 | 183.5544 | −0.0054 |
| MB [Nm] | 41.6755 | 0.4917 | 41.9015 | −0.0504 |

As it is possible to conclude from Tables 4–6, the results obtained by the present model are reliable, based on the convergence of the results for different discretisation, where variations are very small.

### 3.2. Case Studies

The present section examines a set of case studies that focus on truss and frame-type structures made of functionally graded materials whose constituent materials' mixture varies as a function of the height of the structure. It was considered that the thickness of the different members would be much less than their length and that the volume fraction would be associated with the beam's mid-surface. In the latter assumption, this means that when the thickness coincides with the direction of the height of the structure, the volume fraction remains constant.

As the objective of this study is to focus on the linear static and free vibration behaviour of these structures, buckling was not considered at this stage, although we do plan to consider it in future studies. The studies were restricted to aspect ratios within the application domain of the first order shear deformation theory; thus, we restricted the aspect ratio so that it varied between 10 and 20.

### 3.3. Truss-Type Structures

#### 3.3.1. Structure 1

The first case study that we will examine is a simple truss consisting of three members and subject to a concentrated load P at the free node, represented in Figure 7, for which the nodal displacements and the first five natural frequencies of the structure are determined for different values of the power law exponents. These exponents determine the variation profile of the quantity of a given material (metallic or ceramic) within the mixture along the vertical coordinate (Y) of the structure.

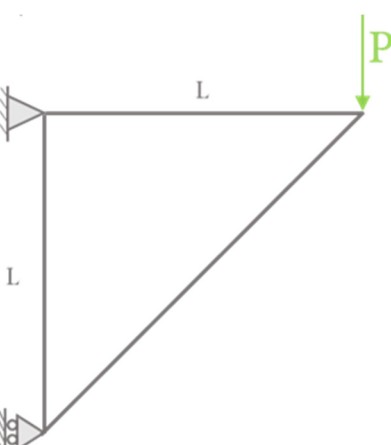

**Figure 7.** Schematic representation of a three-member truss with tip vertical load.

In this case, we considered lengths of 0.2 m for the horizontal and vertical members, with a square cross-section of 20 mm edge, so that the aspect ratio varied between 10 and

15. The coordinate system used can be observed in Figure 8; based on this, the expression of the metal volume fraction was defined as follows:

$$V_f(Y) = \left(\frac{Y}{L} + 1\right)^{ey} \tag{29}$$

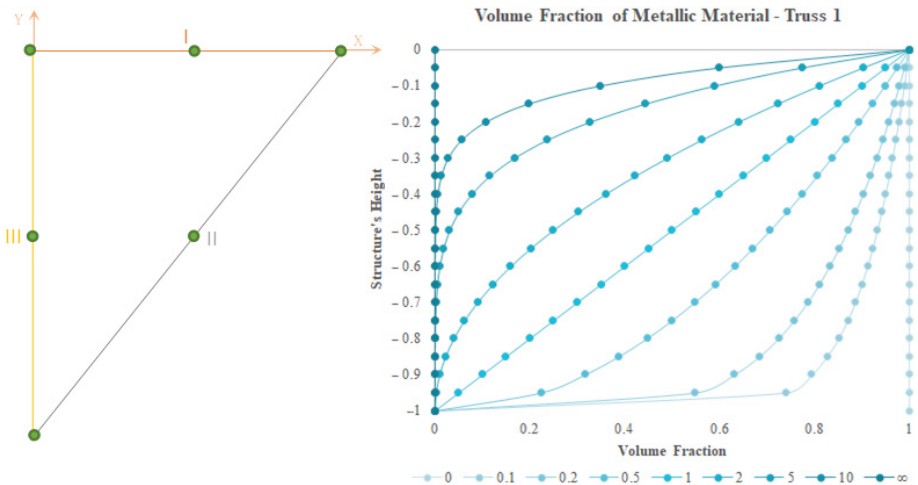

**Figure 8.** Volume fraction evolution through the vertical coordinate of the truss.

Figure 8 schematically presents the minimum discretisation of the structure with three quadratic elements and the evolution of the volume fraction as a function of the vertical Y coordinate.

The first parametric study performed involved an FGM structure (the volume fraction power law of which is described in Equation (29)) that was associated with the variation of the metallic material as a function of the structure's vertical coordinate. The nodal displacements are presented in Table 7, and the first five natural frequencies are shown in Table 8.

**Table 7.** Nodal displacements of the free node of the truss when the volume fraction characterizes the distribution of aluminium.

| ey | Displacement X [μm] | Displacement Y [μm] | Rotation [rad] |
|----|---------------------|---------------------|----------------|
| 0 | 0.7068 | −3.4126 | $1.7857 \times 10^{-5}$ |
| 0.1 | 0.7067 | −3.0564 | $1.6272 \times 10^{-5}$ |
| 0.2 | 0.7066 | −2.8457 | $1.5313 \times 10^{-5}$ |
| 0.5 | 0.7064 | −2.5005 | $1.3665 \times 10^{-5}$ |
| 1 | 0.7064 | −2.2348 | $1.2291 \times 10^{-5}$ |
| 2 | 0.7064 | −2.0099 | $1.1019 \times 10^{-5}$ |
| 5 | 0.7066 | −1.8113 | $9.7844 \times 10^{-6}$ |
| 10 | 0.7067 | −1.7307 | $9.2484 \times 10^{-6}$ |
| ∞ | 0.2474 | −1.1944 | $6.2498 \times 10^{-6}$ |

**Table 8.** First five fundamental frequencies of Truss 1 when the volume fraction characterizes the distribution of aluminium.

| ey | 1st Freq [rad/s] | 2nd Freq [rad/s] | 3rd Freq [rad/s] | 4th Freq [rad/s] | 5th Freq [rad/s] |
|----|------------------|------------------|------------------|------------------|------------------|
| 0 | 5294.9328 | 8221.6085 | 13,113.7575 | 16,576.9632 | 18,223.8911 |
| 0.1 | 5438.4301 | 8427.8559 | 13,605.2249 | 17,017.2024 | 18,786.3044 |
| 0.2 | 5507.1050 | 8560.6395 | 13,844.2795 | 17,291.8409 | 19,071.0106 |
| 0.5 | 5578.4226 | 8790.7384 | 14,097.0130 | 17,707.8291 | 19,501.7499 |
| 1 | 5587.0245 | 8975.0793 | 14,142.9547 | 17,921.0593 | 19,910.5195 |

**Table 8.** *Cont.*

| ey | 1st Freq [rad/s] | 2nd Freq [rad/s] | 3rd Freq [rad/s] | 4th Freq [rad/s] | 5th Freq [rad/s] |
|----|------------------|------------------|------------------|------------------|------------------|
| 2 | 5562.1065 | 9144.7934 | 14,120.0816 | 17,955.3108 | 20,413.3131 |
| 5 | 5528.5888 | 9344.2807 | 14,200.2838 | 17,783.6859 | 21,022.8531 |
| 10 | 5518.4462 | 9461.5509 | 14,353.2407 | 17,606.7775 | 21,292.2856 |
| $\infty$ | 6162.1416 | 9568.1512 | 15,261.5406 | 19,291.9531 | 21,208.6161 |

The properties' evolution (Figure 9a) along the height can be described as follows:

$$P_{FGM}(Y) = P_{Al}V_f + P_{ZrO_2}\left(1 - V_f\right) \tag{30}$$

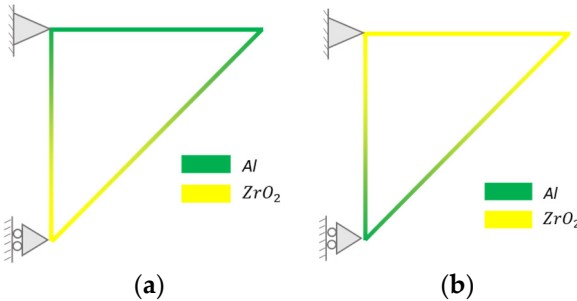

(**a**)     (**b**)

**Figure 9.** Schematic Representation of Truss 1 (constituted by an aluminium–zirconia-graded material along the structure height). (**a**) Volume fraction associated with aluminium; (**b**) volume fraction associated with zirconia.

The results obtained for the displacements of the node under study allowed us to verify that there was a decrease in the vertical displacement with the increase in the power law exponent due to the increasing quantity of material with a higher modulus of elasticity (in this case, ceramic).

On the other hand, if the power law is considered to characterize the distribution of the ceramic material as a function of the vertical coordinate of the structure, a behavioural trend in the opposite sense is verified, as expected. These results are presented in Tables 9 and 10 for the nodal displacements and natural frequencies, respectively.

**Table 9.** Nodal displacements of the free node of the truss when the volume fraction characterizes the distribution of zirconia.

| ey | Displacement X [μm] | Displacement Y [μm] | Rotation [rad] |
|----|---------------------|---------------------|----------------|
| 0 | 0.2474 | −1.1944 | $6.2498 \times 10^{-6}$ |
| 0.1 | 0.2474 | −1.2567 | $6.5296 \times 10^{-6}$ |
| 0.2 | 0.2473 | −1.3201 | $6.8225 \times 10^{-6}$ |
| 0.5 | 0.2472 | −1.5069 | $7.7184 \times 10^{-6}$ |
| 1 | 0.2470 | −1.7694 | $9.0422 \times 10^{-6}$ |
| 2 | 0.2468 | −2.1011 | $1.0823 \times 10^{-5}$ |
| 5 | 0.2467 | −2.4873 | $1.3025 \times 10^{-5}$ |
| 10 | 0.2467 | −2.6730 | $1.4046 \times 10^{-5}$ |
| $\infty$ | 0.7068 | −3.4126 | $1.7857 \times 10^{-5}$ |

The properties' generic distribution (Figure 9b) in this case is now characterized by the following:

$$P_{FGM}(Y) = P_{ZrO_2}V_f + P_{Al}\left(1 - V_f\right) \tag{31}$$

In this case, the nodal displacements presented for a situation in which the power law rules the distribution of the ceramic material along the structure's height exhibits

an antithetical behaviour compared to the previous situation (power law relative to the distribution of the metallic material as a function of the vertical coordinate), meaning that the smaller the amount of ceramic material, the larger are the nodal displacements obtained.

**Table 10.** First five fundamental frequencies of Truss 1 when the volume fraction characterizes the distribution of zirconia.

| ey | 1st Freq [rad/s] | 2nd Freq [rad/s] | 3rd Freq [rad/s] | 4th Freq [rad/s] | 5th Freq [rad/s] |
|----|------------------|------------------|------------------|------------------|------------------|
| 0 | 6162.1416 | 9568.1512 | 15,261.5406 | 19,291.9531 | 21,208.6161 |
| 0.1 | 6106.9814 | 9487.2227 | 15,029.9131 | 19,117.8152 | 20,984.3075 |
| 0.2 | 6059.6603 | 9413.2377 | 14,829.1127 | 18,955.7453 | 20,764.1673 |
| 0.5 | 5964.7921 | 9228.3129 | 14,384.2258 | 18,543.6814 | 20,239.9449 |
| 1 | 5913.5412 | 9014.0362 | 13,997.8667 | 18,124.2110 | 19,789.3659 |
| 2 | 5935.0719 | 8777.9578 | 13,751.7949 | 17,832.1484 | 19,445.8080 |
| 5 | 6013.9751 | 8528.9783 | 13,667.7412 | 17,673.9893 | 19,161.3870 |
| 10 | 6023.1037 | 8386.2242 | 13,633.7039 | 17,541.9697 | 19,104.8654 |
| ∞ | 5294.9328 | 8221.6085 | 13,113.7575 | 16,576.9632 | 18,223.8911 |

From Tables 7–10, several trends were observed, namely, the displacements more directly affected by the load to which Truss 1 is subjected to and a decrease with an increase in the ceramic phase with a higher modulus of elasticity.

As for the displacement along the X direction, it can be observed that when the rule of mixtures is related to the distribution of aluminium (Table 7), it remains constant with the displacement verified to the totally metallic structure, except for the cases wherein the structure is totally ceramic. In the opposite case, when the rule of mixtures is related to the distribution of zirconia (Table 9), it remains constant with the displacement verified for the totally ceramic structure regardless of the power law exponents, except when the structure is totally metallic.

From Table 8, in which the first five natural frequencies of the structure are presented, it is possible to observe that when the volume fraction is associated with the metallic material, the first natural frequency increases from ey = 0 (totally metallic) to ey = 1 and decreases from ey = 1 to ey = 10. For the remaining frequencies, this increase occurs from ey = 0 to ey = 10.

In the case of the volume fraction being directly related to the distribution of the ceramic material (see Table 10 for the natural frequencies), one can observe a decrease in the fundamental frequency from ey = 0 (all-ceramic structure) to ey = 1, followed by an increase from ey = 1 to ey = 10. For the higher-order frequencies, a decrease occurs from ey = 0 to ey = 10. As expected, the lowest results are verified when the structure is totally metallic, and the highest ones are verified when Truss 1 is totally ceramic.

### 3.3.2. Structure 2

In this second case study, we analyse the truss shown in Figure 10 (which consists of nineteen members and is subjected to a concentrated load P at its upper vertex). The displacement of this point and the first five natural frequencies of the structure are presented for different values of the volume fraction law exponent.

In this case, the distance between the supports was considered to be 1.4 m, and the structure's height was considered to be half of that distance; the elements have a square section with a 20 mm side, and the aspect ratio varies between 10 and 20. The origin of the structure coordinate system is located at the left support, as depicted in Figure 11, so the expression of the metal volume fraction is defined by the following:

$$V_f(Y) = \left( \frac{Y}{\frac{7}{2}L} \right)^{ey} \tag{32}$$

Figure 11 schematically presents the minimum discretisation of the structure with 19 quadratic elements and the evolution of the volume fraction as a function of the vertical Y coordinate.

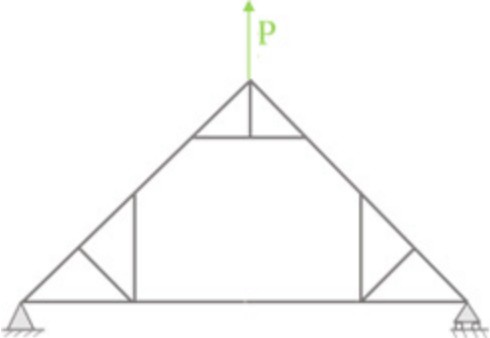

**Figure 10.** Schematic representation of Truss 2 with load P applied to its highest point.

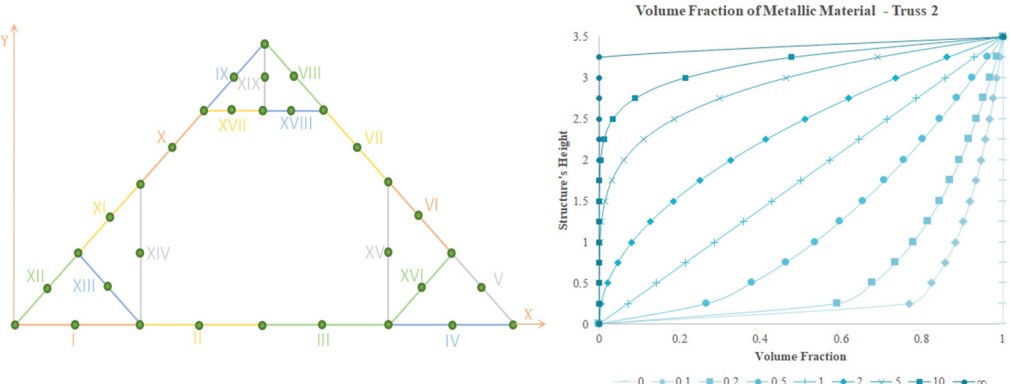

**Figure 11.** Representation of the variation in the volume fraction through the vertical coordinate of Truss 2.

The first parametric study performed considers that the volume fraction described in Equation (32) is directly associated with the variation of the metallic material in the structure as a function of the vertical coordinate of the structure (Figure 12a). The nodal displacements are presented in Table 11, and the natural frequencies are shown in Table 12.

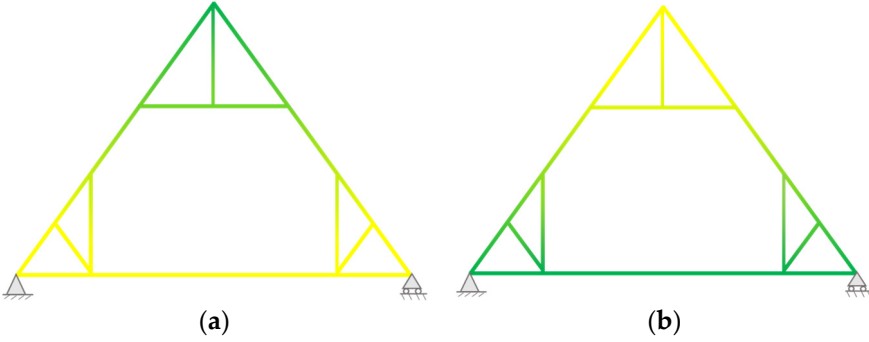

**Figure 12.** Schematic representation of Truss 2. (**a**) Volume fraction characterizing the distribution of metallic material (in this case, aluminium) through the structure's height; (**b**) volume fraction characterizing the distribution of ceramic material (in this case, zirconia) through the structure's height.

The results obtained for the nodal displacements allow us to verify that there is a decrease in the vertical displacement of the selected highest node with the increase in the

power law exponent due to the associated increasing quantity of material with a higher modulus of elasticity (in this case, the ceramic phase).

**Table 11.** Nodal displacements of Truss 2's highest node, where load P is Applied, when the volume fraction characterizes the distribution of aluminium.

| ey | Displacement X [μm] | Displacement Y [μm] | Rotation [rad] |
|----|---------------------|---------------------|----------------|
| 0 | −12.4338 | 47.5739 | $−1.5054 \times 10^{−10}$ |
| 0.1 | −4.3536 | 34.8590 | $3.1781 \times 10^{−10}$ |
| 0.2 | −4.3542 | 32.1102 | $9.2028 \times 10^{−10}$ |
| 0.5 | −4.3542 | 27.6032 | $3.5770 \times 10^{−10}$ |
| 1 | −4.3537 | 24.1793 | $2.8081 \times 10^{−10}$ |
| 2 | −4.3533 | 21.3359 | $5.5376 \times 10^{−10}$ |
| 5 | −4.3530 | 18.8600 | $4.5957 \times 10^{−10}$ |
| 10 | −4.3526 | 17.8269 | $−5.6386 \times 10^{−10}$ |
| ∞ | −4.3518 | 16.6509 | $1.3354 \times 10^{−9}$ |

**Table 12.** First five fundamental frequencies of Truss 2 when the volume fraction characterizes the distribution of aluminium.

| ey | 1st Freq [rad/s] | 2nd Freq [rad/s] | 3rd Freq [rad/s] | 4th Freq [rad/s] | 5th Freq [rad/s] |
|----|------------------|------------------|------------------|------------------|------------------|
| 0 | 388.5431 | 601.1494 | 1271.0603 | 1446.3974 | 2682.3516 |
| 0.1 | 479.6393 | 539.8995 | 1452.5925 | 1477.5564 | 3035.4102 |
| 0.2 | 475.9038 | 553.0215 | 1478.6352 | 1481.3936 | 3073.2990 |
| 0.5 | 466.6954 | 582.9778 | 1471.2405 | 1543.1080 | 3114.4485 |
| 1 | 459.0512 | 615.9931 | 1461.9136 | 1603.7148 | 3125.3727 |
| 2 | 454.7860 | 652.1872 | 1459.2405 | 1657.5705 | 3123.6320 |
| 5 | 453.7501 | 686.3469 | 1469.5407 | 1686.4211 | 3121.5422 |
| 10 | 452.7845 | 696.2437 | 1476.3244 | 1683.4642 | 3121.6488 |
| ∞ | 452.1537 | 699.5999 | 1479.2326 | 1683.2820 | 3121.6682 |

On the other hand, if it is considered that the power law characterizes the distribution of the ceramic material as a function of the vertical coordinate (Figure 12b), an inverse behaviour to the one considered in the previous study is verified. The results of nodal displacements and natural frequencies are presented in Tables 13 and 14, respectively, for this configuration.

**Table 13.** Nodal displacements of Truss 2's highest node, where load P is applied, when the volume fraction characterizes the distribution of zirconia.

| ey | Displacement X [μm] | Displacement Y [μm] | Rotation [rad] |
|----|---------------------|---------------------|----------------|
| 0 | −4.3518 | 16.6509 | $1.3354 \times 10^{−9}$ |
| 0.1 | −12.4001 | 25.5124 | $1.2985 \times 10^{−9}$ |
| 0.2 | −12.4047 | 26.3631 | $2.2936 \times 10^{−10}$ |
| 0.5 | −12.4146 | 28.8642 | $3.8254 \times 10^{−10}$ |
| 1 | −12.4238 | 32.2969 | $1.6543 \times 10^{−9}$ |
| 2 | −12.4300 | 36.6319 | $1.5049 \times 10^{−9}$ |
| 5 | −12.4311 | 41.7767 | $−1.7748 \times 10^{−9}$ |
| 10 | −12.4318 | 44.3473 | $9.5548 \times 10^{−10}$ |
| ∞ | −12.4338 | 47.5739 | $−1.5054 \times 10^{−10}$ |

As can be observed in Table 13, where the volume fraction is directly related to the distribution of the ceramic material along the structure' height, one can conclude that, as expected, the smaller the amount of ceramic material whose modulus of elasticity is higher than that of the metallic material, the larger are the nodal displacements obtained.

From Tables 11–14, several trends can be observed. In concrete, one can observe that the displacements more visibly affected by the load to which Truss 2 is subjected to (displacement along y direction) decrease with the increase in the ceramic phase.

**Table 14.** First five Fundamental frequencies of Truss 2 when the volume fraction characterizes the distribution of zirconia.

| ey | 1st Freq [rad/s] | 2nd Freq [rad/s] | 3rd Freq [rad/s] | 4th Freq [rad/s] | 5th Freq [rad/s] |
|---|---|---|---|---|---|
| 0 | 452.1537 | 699.5999 | 1479.2326 | 1683.2820 | 3121.6682 |
| 0.1 | 385.3497 | 755.9448 | 1289.1583 | 1692.1243 | 2351.6669 |
| 0.2 | 387.7700 | 750.6037 | 1296.4966 | 1683.8258 | 2381.0814 |
| 0.5 | 392.0137 | 734.2695 | 1310.2818 | 1656.6153 | 2437.5914 |
| 1 | 394.3011 | 710.9035 | 1320.3630 | 1612.3869 | 2484.6958 |
| 2 | 393.0573 | 675.8203 | 1318.4429 | 1543.3509 | 2534.9549 |
| 5 | 388.0187 | 626.1564 | 1291.4686 | 1461.8514 | 2625.8542 |
| 10 | 387.6009 | 606.9311 | 1276.1017 | 1445.8725 | 2674.5813 |
| ∞ | 388.5431 | 601.1494 | 1271.0603 | 1446.3974 | 2682.3516 |

As for the displacement along the X direction and when the volume fraction is directly related to the distribution of aluminium (Table 11), this displacement remains equal to the one presented from the totally ceramic structure, regardless of the power law exponent. In the opposite case, when the volume fraction is related to the distribution of zirconia (Table 13), the displacement remains equal to the one presented by the metallic structure despite the power law exponent values. As for the rotation, there were no clear trends to highlight.

When the volume fraction characterizes the distribution of the metallic material phase, it is also possible to conclude from Table 12 that the first natural frequency decreases from ey = 0.1 to ey = 10; for the fourth frequency, the maximum value is obtained when ey = 5, and for the remaining ones, the maximum value is obtained when ey = 10. The highest value for the first frequency is obtained when this truss has a power law exponent that is equal to 0.1, and for the remaining frequencies, the highest values are obtained when truss 2 is totally ceramic. As expected, this structure presents the lowest fundamental frequency when it is fully metallic.

In the opposite situation, when the power law volume fraction characterises the distribution of the ceramic material, the results of the natural frequencies depicted in Table 14 show an increasing trend from ey = 0.1 to ey = 1 and a decreasing one from ey = 1 to ey = 10. For the remaining ones, there is a decrease from ey = 0.1 to ey = ∞. Also, as expected, the lowest frequencies are obtained when the structure is completely metallic. The highest values for the first, third, and fifth frequencies are obtained when truss 2 is totally ceramic, but for the remaining ones, the highest values are obtained when ey = 0.1.

### 3.4. Frame-Type Structures

3.4.1. Structure 3

The third case study is a simple frame consisting of three members that is subjected to a distributed load in the transverse direction of the upper horizontal member, which is additionally subjected to a concentrated load P along its length, as presented in Figure 13.

The upper nodal displacements and the first five natural frequencies of the structure are presented in Tables 15–18, as for the previous case studies.

**Table 15.** Nodal displacements of Frame 1's upper nodes when the volume fraction characterizes the distribution of aluminium.

| ey | Upper-Left Node | | | Upper-Right Node | | |
|---|---|---|---|---|---|---|
| | Displacement X [mm] | Displacement Y [mm] | Rotation [rad] | Displacement X [mm] | Displacement Y [mm] | Rotation [rad] |
| 0 | 0.1792 | −0.0008 | $3.5491 \times 10^{-4}$ | 0.1790 | −0.0021 | $-3.6894 \times 10^{-6}$ |

**Table 15.** *Cont.*

| ey | Upper-Left Node | | | Upper-Right Node | | |
|---|---|---|---|---|---|---|
| | Displacement X [mm] | Displacement Y [mm] | Rotation [rad] | Displacement X [mm] | Displacement Y [mm] | Rotation [rad] |
| 0.1 | 0.1487 | −0.0007 | $3.3344 \times 10^{-4}$ | 0.1485 | −0.0018 | $-1.0967 \times 10^{-5}$ |
| 0.2 | 0.1321 | −0.0007 | $3.2006 \times 10^{-4}$ | 0.1318 | −0.0016 | $-1.3568 \times 10^{-5}$ |
| 0.5 | 0.1097 | −0.0006 | $2.9831 \times 10^{-4}$ | 0.1094 | −0.0013 | $-1.3790 \times 10^{-5}$ |
| 1 | 0.0978 | −0.0005 | $2.8233 \times 10^{-4}$ | 0.0975 | −0.0011 | $-9.1097 \times 10^{-6}$ |
| 2 | 0.0913 | −0.0004 | $2.6917 \times 10^{-4}$ | 0.0911 | −0.0009 | $1.2487 \times 10^{-6}$ |
| 5 | 0.0871 | −0.0004 | $2.5687 \times 10^{-4}$ | 0.0869 | −0.0008 | $2.1297 \times 10^{-5}$ |
| 10 | 0.0848 | −0.0003 | $2.5096 \times 10^{-4}$ | 0.0846 | −0.0007 | $3.5145 \times 10^{-5}$ |
| ∞ | 0.0627 | −0.0003 | $1.2422 \times 10^{-4}$ | 0.0626 | −0.0007 | $-1.2913 \times 10^{-6}$ |

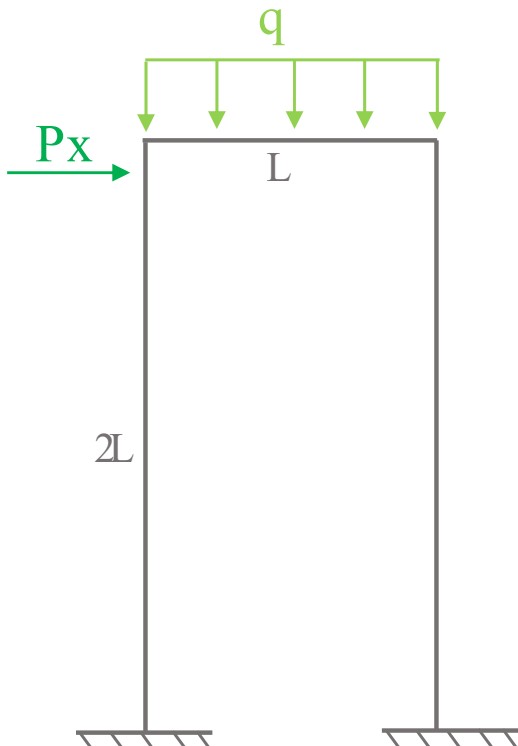

**Figure 13.** Schematic representation of Frame 1 and the loading considered.

**Table 16.** First five fundamental frequencies of Frame 1 when the volume fraction characterizes the distribution of aluminium.

| ey | 1st Freq [rad/s] | 2nd Freq [rad/s] | 3rd Freq [rad/s] | 4th Freq [rad/s] | 5th Freq [rad/s] |
|---|---|---|---|---|---|
| 0 | 739.8785 | 3219.5155 | 4279.3392 | 8288.5159 | 10,736.1255 |
| 0.1 | 809.2904 | 3393.0264 | 4499.5592 | 8559.9600 | 11,183.8027 |
| 0.2 | 851.7884 | 3479.9698 | 4612.8459 | 8705.0707 | 11,414.6427 |
| 0.5 | 917.1649 | 3571.5958 | 4748.1336 | 8913.3232 | 11,710.3824 |
| 1 | 950.5337 | 3572.8360 | 4793.6135 | 9055.3907 | 11,861.1267 |
| 2 | 952.1689 | 3525.5616 | 4808.4520 | 9199.6831 | 11,978.8475 |
| 5 | 926.5586 | 3479.8271 | 4839.8815 | 9423.2691 | 12,155.8452 |
| 10 | 910.8577 | 3468.6017 | 4855.0349 | 9585.1023 | 12,259.0905 |
| ∞ | 861.0126 | 3746.8095 | 4980.2095 | 9646.0159 | 12,494.4966 |

The characteristics of the cross-section are equal to those of the previous case; the distance between the supports is 0.2 m, and the height of the structure was set so that

it could be double that distance. The origin of the coordinate system coincides with the left-hand side support. The metal volume fraction is in this case defined as follows:

$$V_f(Y) = \left(\frac{Y}{2L}\right)^{ey} \tag{33}$$

Figure 14 schematically presents the minimum discretisation of the structure with 19 quadratic beam elements and the evolution of the volume fraction as a function of the vertical $Y$ coordinate.

**Table 17.** Nodal displacements of Frame 1's upper nodes when the volume fraction characterizes the distribution of zirconia.

| ey | Displacement X [mm] | Displacement Y [mm] | Rotation [rad] | Displacement X [mm] | Displacement Y [mm] | Rotation [rad] |
|---|---|---|---|---|---|---|
| 0 | 0.0627 | −0.0003 | $1.2422 \times 10^{-4}$ | 0.0626 | −0.0007 | $-1.2879 \times 10^{-6}$ |
| 0.1 | 0.0679 | −0.0003 | $1.2761 \times 10^{-4}$ | 0.0679 | −0.0008 | $1.4256 \times 10^{-7}$ |
| 0.2 | 0.0730 | −0.0003 | $1.3076 \times 10^{-4}$ | 0.0729 | −0.0008 | $1.4870 \times 10^{-6}$ |
| 0.5 | 0.0855 | −0.0003 | $1.3833 \times 10^{-4}$ | 0.0855 | −0.0010 | $4.5551 \times 10^{-6}$ |
| 1 | 0.0978 | −0.0004 | $1.4551 \times 10^{-4}$ | 0.0977 | −0.0013 | $6.4263 \times 10^{-6}$ |
| 2 | 0.1081 | −0.0004 | $1.5131 \times 10^{-4}$ | 0.1080 | −0.0015 | $4.8509 \times 10^{-6}$ |
| 5 | 0.1205 | −0.0005 | $1.5623 \times 10^{-4}$ | 0.1205 | −0.0018 | $-3.4421 \times 10^{-6}$ |
| 10 | 0.1315 | −0.0006 | $1.5870 \times 10^{-4}$ | 0.1314 | −0.0020 | $-1.0787 \times 10^{-5}$ |
| ∞ | 0.1792 | −0.0008 | $3.5490 \times 10^{-4}$ | 0.1790 | −0.0021 | $-3.6975 \times 10^{-6}$ |

**Table 18.** First five fundamental frequencies of Frame 1 when the volume fraction characterizes the distribution of zirconia.

| ey | 1st Freq [rad/s] | 2nd Freq [rad/s] | 3rd Freq [rad/s] | 4th Freq [rad/s] | 5th Freq [rad/s] |
|---|---|---|---|---|---|
| 0 | 861.0126 | 3746.8095 | 4980.2095 | 9646.0159 | 12,494.4966 |
| 0.1 | 826.0344 | 3665.2790 | 4878.3129 | 9521.6613 | 12,306.1368 |
| 0.2 | 796.5309 | 3600.3670 | 4796.4566 | 9416.8544 | 12,154.7020 |
| 0.5 | 737.4858 | 3489.6473 | 4651.1542 | 9203.3853 | 11,874.9982 |
| 1 | 697.4650 | 3453.9136 | 4585.9785 | 9031.6921 | 11,700.2361 |
| 2 | 679.0346 | 3495.7345 | 4593.5288 | 8856.6486 | 11,561.3434 |
| 5 | 670.7692 | 3587.1460 | 4604.7033 | 8567.1039 | 11,352.2676 |
| 10 | 662.8164 | 3624.3249 | 4552.2071 | 8373.0567 | 11,274.0785 |
| ∞ | 739.8785 | 3219.5155 | 4279.3392 | 8288.5159 | 10,736.1255 |

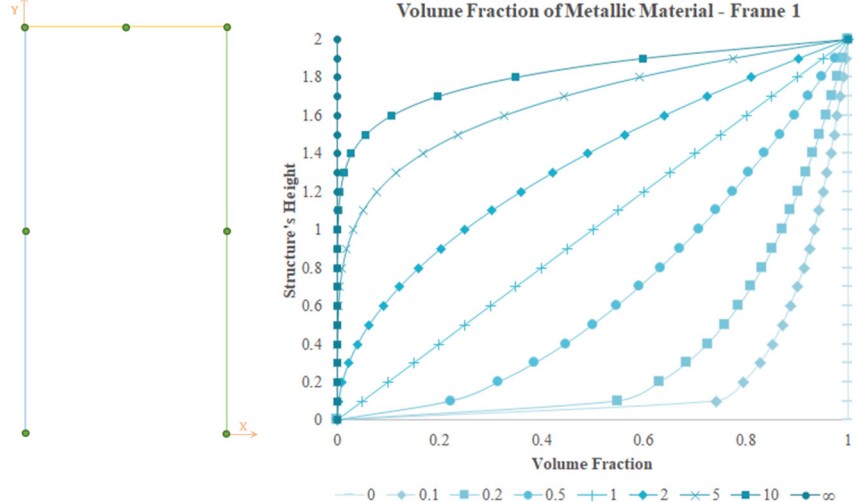

**Figure 14.** Representation of the variation in the volume fraction through the vertical coordinate of Frame1.

In the first parametric study, we considered the volume fraction from Equation (33) to be directly associated with the variation in the metallic material in the structure as a function of the vertical coordinate of the structure (Figure 15a). The nodal displacements achieved are presented in Table 15, and the natural frequencies can be observed in Table 16.

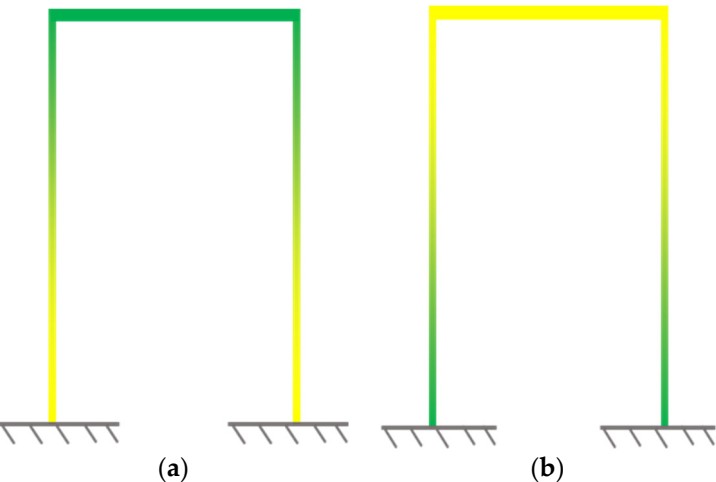

**Figure 15.** Schematic representation of Frame 1. (**a**) Volume fraction characterizes the distribution of metallic material (aluminium) through the structure's height; (**b**) volume fraction characterizes the distribution of ceramic material (zirconia) through the structure's height.

The results obtained for the nodal displacements allowed us to conclude that there is a decrease in the vertical displacement of the nodes under study with an increase in the power law exponent due to the increasing quantity of the material with a higher modulus of elasticity (in this case, ceramic) (Figure 15b).

Again, if the volume fraction characterizes the distribution of the ceramic material as a function of the vertical coordinate of the structure (Figure 15b), an expected opposite behaviour is shown (Tables 18 and 19).

**Table 19.** Nodal displacements of Frame 2's upper nodes when the volume fraction characterizes the distribution of aluminium.

| ey | Upper-Left Node | | | Upper-Right Node | | |
|---|---|---|---|---|---|---|
| | Displacement X [μm] | Displacement Y [μm] | Rotation [rad] | Displacement X [μm] | Displacement Y [μm] | Rotation [rad] |
| 0 | 2.9111 | −0.6235 | $2.5819 \times 10^{-5}$ | 2.7199 | −1.3642 | $-1.4635 \times 10^{-5}$ |
| 0.1 | 2.4922 | −0.5424 | $2.4541 \times 10^{-5}$ | 2.3003 | −1.1286 | $-1.4954 \times 10^{-5}$ |
| 0.2 | 2.2530 | −0.4939 | $2.3730 \times 10^{-5}$ | 2.0605 | −0.9975 | $-1.5014 \times 10^{-5}$ |
| 0.5 | 1.8798 | −0.4144 | $2.2257 \times 10^{-5}$ | 1.6863 | −0.8008 | $-1.4818 \times 10^{-5}$ |
| 1 | 1.6155 | −0.3539 | $2.0917 \times 10^{-5}$ | 1.4210 | −0.6706 | $-1.4300 \times 10^{-5}$ |
| 2 | 1.4137 | −0.3035 | $1.9475 \times 10^{-5}$ | 1.2180 | −0.5802 | $-1.3434 \times 10^{-5}$ |
| 5 | 1.2563 | −0.2592 | $1.7585 \times 10^{-5}$ | 1.0593 | −0.5193 | $-1.1986 \times 10^{-5}$ |
| 10 | 1.1977 | −0.2413 | $1.6344 \times 10^{-5}$ | 0.9999 | −0.4996 | $-1.0900 \times 10^{-5}$ |
| ∞ | 1.0189 | −0.2182 | $9.0367 \times 10^{-6}$ | 0.9520 | −0.4775 | $-5.1221 \times 10^{-5}$ |

From Table 17, it is possible to see that the smaller the amount of ceramic material, whose modulus of elasticity, in the present case, is higher than that of the metallic material, the larger are the nodal displacements obtained.

From Tables 15–18, several trends can be observed, namely, that displacements decrease with the increase in the ceramic phase characterized by a higher modulus of elasticity when compared to the that of the metallic phase.

For the Frame 1 structure, the first five natural frequencies of which are presented in Table 16, one can conclude that when the power law volume fraction is associated with the metallic material, the first natural frequency increases from ey = 0 to ey = 2 and decreases from ey = 2 to ey = ∞. For the second frequency, the values increase from ey = 0 to ey = 1 and decrease from ey = 1 to ey = 10, being higher when ey = ∞ due to the complete ceramic character of the structure. For the remaining frequencies, one can observe that there is an increasing trend from ey = 0 to ey = ∞.

In the case of the power law characterising the distribution of the ceramic material, the results presented in Table 18 show that, for the first and second frequencies, there is a decreasing trend (from ey = 0 to ey = 10). The third frequency decreases from ey = 0 to ey = 1 and increases from ey = 1 to ey = 5 before decreasing. The remaining higher-order frequencies decrease from ey = 0 to ey = ∞.

### 3.4.2. Structure 4

The fourth case study focused on the frame-type structure represented in Figure 16, which is consists of 19 members and is subjected to a horizontal concentrated load P at the upper-left node and to a transverse uniformly distributed load in the upper-horizontal member.

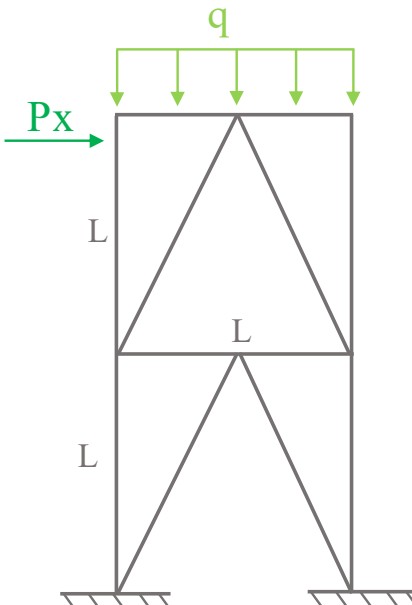

**Figure 16.** Schematic representation of Frame 2 and the loading considered.

A parametric study concerning the influence of the power law exponent was performed to assess the upper nodal displacements and the first five natural frequencies of the structure.

The distance between the structure's supports and the cross-section is equal to the previous case study. The structure's height is twice that distance. The metal volume fraction for this structure, considering the coordinate system origin shown in Figure 17, is defined by the following:

$$V_f(Y) = \left( \frac{Y}{2L} \right)^{ey} \tag{34}$$

Figure 17 schematically presents the minimum discretisation of the structure with 19 quadratic elements and the evolution of the volume fraction as a function of the vertical *Y* structure coordinate.

In Figure 18, one can observe a schematic representation of the graded material evolution, considering the writing of the rule of mixtures homogenisation scheme.

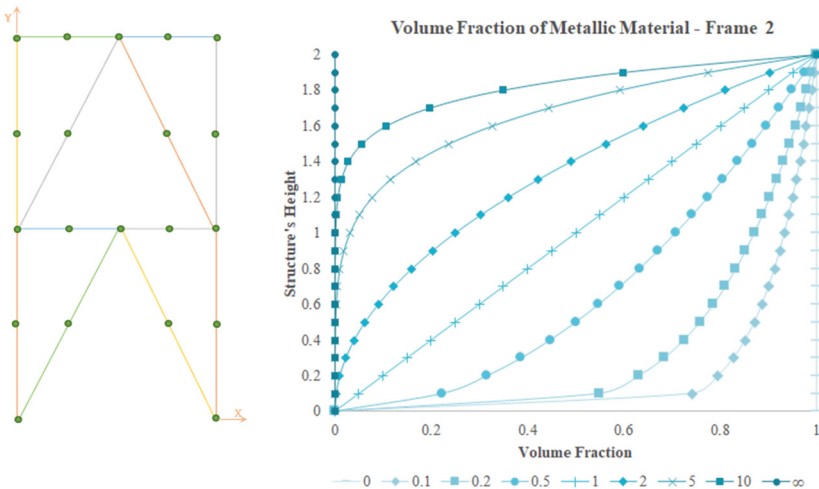

**Figure 17.** Representation of the variation in the volume fraction through the vertical coordinate of the truss considered.

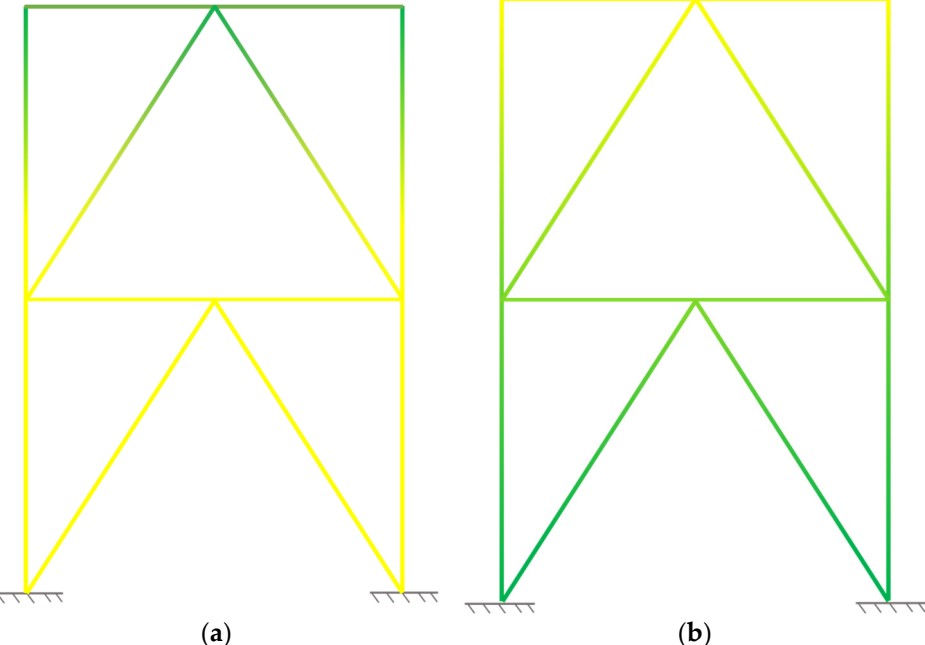

**Figure 18.** Schematic representation of Frame 2 considering that the rule of mixtures is such that (**a**) the volume fraction characterizes the distribution of metallic material (in this case, aluminium) through the structure's height or (**b**) the volume fraction characterizes the distribution of ceramic material (in this case, zirconia) through the structure's height.

The results obtained for the nodal displacements (presented in Table 19) denote a decrease in the vertical displacement of the upper nodes with the increase in the volume fraction law exponent due to the increasing quantity of ceramic material (with a higher modulus of elasticity). In Table 20, the first five fundamental frequencies of Frame 2 are presented, with the volume fraction expression related to the metallic material phase.

Again, if one considers the homogenisation scheme written in the opposite direction, that is, the volume fraction law characterising the distribution of the ceramic material as a function of the vertical coordinate of the structure, an inverted behaviour is found as expected and as observed in Tables 21 and 22.

From Tables 19 and 21, the same trends that occurred for Frame 1 can be observed. It is clear that the existence of the intermediate horizontal beam on Frame 2 yields

a significant reduction in the nodal displacements and a remarkable increase in the fundamental frequencies.

**Table 20.** First five fundamental frequencies of Frame 2 when the volume fraction characterizes the distribution of aluminium.

| ey | 1st Freq [rad/s] | 2nd Freq [rad/s] | 3rd Freq [rad/s] | 4th Freq [rad/s] | 5th Freq [rad/s] |
|---|---|---|---|---|---|
| 0 | 4445.2573 | 9725.9235 | 10,329.8695 | 10,577.9434 | 11,776.9889 |
| 0.1 | 4811.1142 | 10,414.2152 | 10,526.8269 | 10,819.6200 | 12,275.9442 |
| 0.2 | 5048.3392 | 10,653.8680 | 10,764.4493 | 10,991.9750 | 12,510.2352 |
| 0.5 | 5450.4551 | 10,878.4043 | 11,214.4145 | 11,293.3519 | 12,908.1692 |
| 1 | 5715.0803 | 11,040.2213 | 11,421.5484 | 11,524.6531 | 13,214.8139 |
| 2 | 5813.1618 | 11,120.2976 | 11,480.3985 | 11,734.8041 | 13,340.6033 |
| 5 | 5685.5721 | 11,073.6606 | 11,530.2708 | 11,772.6205 | 13,345.3817 |
| 10 | 5554.7988 | 11,048.3905 | 11,530.2312 | 11,696.3605 | 13,222.3765 |
| ∞ | 5173.3015 | 11,318.8428 | 12,021.7047 | 12,310.4084 | 13,705.8341 |

**Table 21.** Nodal displacements of Frame 2's upper nodes when the volume fraction characterizes the distribution of zirconia.

| ey | Upper-Left Node | | | Upper-Right Node | | |
|---|---|---|---|---|---|---|
| | Displacement X [μm] | Displacement Y [μm] | Rotation [rad] | Displacement X [μm] | Displacement Y [μm] | Rotation [rad] |
| 0 | 1.0189 | −0.2182 | $9.0367 \times 10^{-6}$ | 0.9520 | −0.4775 | $-5.1221 \times 10^{-6}$ |
| 0.1 | 1.0942 | −0.2323 | $9.2535 \times 10^{-6}$ | 1.0274 | −0.5204 | $-5.0442 \times 10^{-6}$ |
| 0.2 | 1.1719 | −0.2468 | $9.4754 \times 10^{-6}$ | 1.1052 | −0.5646 | $-4.9613 \times 10^{-6}$ |
| 0.5 | 1.3997 | −0.2890 | $1.0125 \times 10^{-5}$ | 1.3333 | −0.6941 | $-4.7161 \times 10^{-6}$ |
| 1 | 1.7078 | −0.3472 | $1.1026 \times 10^{-5}$ | 1.6417 | −0.8675 | $-4.4243 \times 10^{-6}$ |
| 2 | 2.0795 | −0.4238 | $1.2214 \times 10^{-5}$ | 2.0137 | −1.0699 | $-4.2611 \times 10^{-6}$ |
| 5 | 2.4596 | −0.5186 | $1.3793 \times 10^{-5}$ | 2.3939 | −1.2512 | $-4.7590 \times 10^{-6}$ |
| 10 | 2.6079 | −0.5633 | $1.4695 \times 10^{-5}$ | 2.5424 | −1.3060 | $-5.4712 \times 10^{-6}$ |
| ∞ | 2.9111 | −0.6235 | $2.5819 \times 10^{-5}$ | 2.7199 | −1.3642 | $-1.4635 \times 10^{-5}$ |

**Table 22.** First five fundamental frequencies of Frame 2 when the volume fraction characterizes the distribution of zirconia.

| ey | 1st Freq [rad/s] | 2nd Freq [rad/s] | 3rd Freq [rad/s] | 4th Freq [rad/s] | 5th Freq [rad/s] |
|---|---|---|---|---|---|
| 0 | 5173.3015 | 11,318.8428 | 12,021.7047 | 12,310.4084 | 13,705.8341 |
| 0.1 | 4992.7154 | 10,995.0986 | 11,948.2082 | 12,223.8979 | 13,381.9534 |
| 0.2 | 4828.3929 | 10,717.5537 | 11,880.4385 | 12,147.0500 | 13,116.0356 |
| 0.5 | 4446.2368 | 10,129.7748 | 11,710.5742 | 11,972.3095 | 12,585.0147 |
| 1 | 4097.5458 | 9660.0454 | 11,429.7611 | 11,796.9219 | 11,901.6716 |
| 2 | 3850.6378 | 9374.3909 | 10,941.9646 | 11,576.1461 | 11,621.1784 |
| 5 | 3786.5223 | 9286.4295 | 10,806.4475 | 11,335.5703 | 11,515.7923 |
| 10 | 3856.4066 | 9286.2461 | 11,013.5120 | 11,375.3596 | 11,549.5688 |
| ∞ | 4445.2573 | 9725.9235 | 10,329.8695 | 10,577.9434 | 11,776.9889 |

From Table 20, the results obtained for the first five natural frequencies of the structure when the power law is relative to the metallic material show that, for the first, second, and fifth natural frequencies, they increase from *ey* = 0 to ey = 2 and decrease from ey = 2 to ey = 10. For the remaining frequencies, there is an increasing trend from ey = 0 to ey = 5 before they decrease from ey = 5 to ey = 10. When ey = ∞, which means the structure is fully ceramic, the highest frequencies are observed.

When the power law volume fraction characterises the distribution of the ceramic material (for which Table 22 shows all of the first five frequencies), except the for the fourth

frequency, a decreasing trend can be observed from ey = 0 to ey = 5, and an increase from ey = 5 to ey = ∞ can also be observed. For the fourth frequency, there is a decrease from ey = 0 to ey = ∞.

### 3.5. Differently Oriented Cross-Section Areas with the Same Value

This last study was conducted to compare the structural response of the first frame-type structure when different aspect ratio cross-sections are considered. For this study, we maintained the value of the cross-sectional area but changed the relationship between the sides. Initially, a square cross-section was considered, but we subsequently considered that one of the sides would have four times the dimension of the other. Figure 19 schematically represents the three sections analysed. This study enabled us to assess the influence that the second area moment has on the behaviour of the FGM structure. In this case, when the section is changed from A to B, the second moment increases four times; when the section is changed from A to C, the second moment decreases four times.

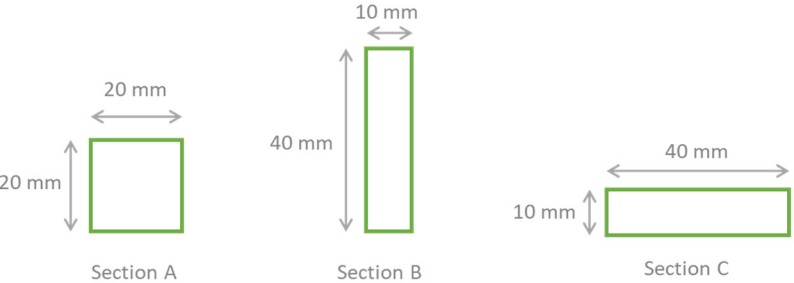

**Figure 19.** Cross-sections considered with the same cross-section area.

A comparison between the most significant nodal displacements is presented in Table 23, where the displacement in the X direction and the rotation of the upper-left node are displayed for each section. It is important to note that, in this case, the aspect ratio for the section B situation varies between 5 and 10; for the section C situation, it varies between 20 and 40.

**Table 23.** Nodal displacements of Frame 1's upper-left node when the volume fraction characterizes the distribution of aluminium for each cross-section.

| ey | Section A | | Section B | | Section C | |
|---|---|---|---|---|---|---|
| | Disp. $X$ [mm] | Rotation [rad] | Disp. $X$ [mm] | Rotation [rad] | Disp. $X$ [mm] | Rotation [rad] |
| 0 | 0.1792 | $3.5501 \times 10^{-4}$ | 0.0473 | $9.7169 \times 10^{-5}$ | 0.7066 | $1.3851 \times 10^{-3}$ |
| 0.1 | 0.1510 | $3.3482 \times 10^{-4}$ | 0.0398 | $9.0912 \times 10^{-5}$ | 0.5952 | $1.3090 \times 10^{-3}$ |
| 0.2 | 0.1360 | $3.2243 \times 10^{-4}$ | 0.0358 | $8.7192 \times 10^{-5}$ | 0.5365 | $1.2628 \times 10^{-3}$ |
| 0.5 | 0.1149 | $3.0202 \times 10^{-4}$ | 0.0302 | $8.1093 \times 10^{-5}$ | 0.4535 | $1.1852 \times 10^{-3}$ |
| 1 | 0.1024 | $2.8639 \times 10^{-4}$ | 0.0269 | $7.6525 \times 10^{-5}$ | 0.4043 | $1.1254 \times 10^{-3}$ |
| 2 | 0.0949 | $2.7336 \times 10^{-4}$ | 0.0249 | $7.2824 \times 10^{-5}$ | 0.3750 | $1.0750 \times 10^{-3}$ |
| 5 | 0.0898 | $2.6083 \times 10^{-4}$ | 0.0235 | $6.9462 \times 10^{-5}$ | 0.3546 | $1.0258 \times 10^{-3}$ |
| 10 | 0.0867 | $2.5397 \times 10^{-4}$ | 0.0228 | $6.7735 \times 10^{-5}$ | 0.3423 | $9.9841 \times 10^{-4}$ |
| ∞ | 0.0627 | $1.2422 \times 10^{-4}$ | 0.0166 | $3.4010 \times 10^{-5}$ | 0.2474 | $4.8485 \times 10^{-4}$ |

For each section considered, the displacements visibly vary depending on the exponent of the power law, confirming a significant influence of the second area moment.

From Table 23, it is clear that when the section is changed from A to B, which means that the section's height doubles and the width halves, the displacements evaluated decrease by a factor between 3.7 and 3.8. On the other hand, if the section is changed from A to C, which means that section's height halves and the width is doubled, the displacements evaluated increase by a factor of between 3.9 and 4.

The fundamental frequencies of this structure for each cross-section are next presented in Table 24.

**Table 24.** Fundamental Frequencies for Frame 1 structures for each cross-section.

| ey | 1st Freq [rad/s] | | |
|---|---|---|---|
| | Section A | Section B | Section C |
| 0 | 739.7862 | 1446.3012 | 372.1516 |
| 0.1 | 808.9194 | 1583.0218 | 406.9964 |
| 0.2 | 851.7885 | 1667.0039 | 428.3063 |
| 0.5 | 917.1649 | 1796.9389 | 461.0486 |
| 1 | 950.5337 | 1864.3238 | 477.6859 |
| 2 | 952.1689 | 1869.0025 | 478.4118 |
| 5 | 926.5586 | 1818.2592 | 465.5766 |
| 10 | 910.8577 | 1785.7504 | 457.8059 |
| ∞ | 861.0126 | 1683.1604 | 433.0391 |

From Table 24, it can be observed that when the section is changed from A to B, the fundamental frequencies double; on the other hand, if the section is changed from A to C, they are halved. We verified the expected trends of both situations. Finally, the nodal generalized displacements of the central node of the horizontal element are presented in Tables 25–27, according to each section considered.

**Table 25.** Displacement according to the global x direction of the horizontal member's central node.

| Ey | Displacement $X$ [mm] | | |
|---|---|---|---|
| | Section A | Section B | Section C |
| 0 | 0.1790 | 0.0471 | 0.7064 |
| 0.1 | 0.1507 | 0.0396 | 0.5950 |
| 0.2 | 0.1358 | 0.0356 | 0.5363 |
| 0.5 | 0.1147 | 0.0300 | 0.4533 |
| 1 | 0.1022 | 0.0267 | 0.4041 |
| 2 | 0.0947 | 0.0246 | 0.3747 |
| 5 | 0.0895 | 0.0233 | 0.3544 |
| 10 | 0.0865 | 0.0225 | 0.3421 |
| ∞ | 0.0626 | 0.0165 | 0.2473 |

**Table 26.** Displacement according to the global y direction of the horizontal member's central node.

| ey | Displacement $Y$ [µm] | | |
|---|---|---|---|
| | Section A | Section B | Section C |
| 0 | −4.2383 | −2.6961 | −10.3540 |
| 0.1 | −4.0167 | −2.4286 | −10.3191 |
| 0.2 | −3.8685 | −2.2707 | −10.2358 |
| 0.5 | −3.5959 | −2.0090 | −9.9229 |
| 1 | −3.3197 | −1.7993 | −9.3830 |
| 2 | −2.9714 | −1.6008 | −8.4372 |
| 5 | −2.4329 | −1.3749 | −6.6493 |
| 10 | −2.0565 | −1.2447 | −5.2874 |
| ∞ | −1.4818 | −0.9436 | −3.6228 |

From Table 25, one can observe the same behaviour already observed in Tables 26 and 27. When the section is altered from A to B, the displacements and rotation evaluated decrease significantly; if section A is altered to the C configuration, the displacements and rotation evaluated experience a significant increase. From Table 25, it is clear that the evaluated central node displacement decreases by a factor of approximately

3.8 if the section is altered from A to B, while if the section is altered from A to C, the evaluated displacement increases by a factor of approximately 3.95.

**Table 27.** Rotation of horizontal member's central node.

| ey | Rotation [rad] | | |
|---|---|---|---|
| | **Section A** | **Section B** | **Section C** |
| 0 | $-1.2536 \times 10^{-4}$ | $-2.3322 \times 10^{-5}$ | $-5.3261 \times 10^{-4}$ |
| 0.1 | $-1.2650 \times 10^{-4}$ | $-2.4849 \times 10^{-5}$ | $-5.3212 \times 10^{-4}$ |
| 0.2 | $-1.2626 \times 10^{-4}$ | $-2.5546 \times 10^{-5}$ | $-5.2967 \times 10^{-4}$ |
| 0.5 | $-1.2484 \times 10^{-4}$ | $-2.6189 \times 10^{-5}$ | $-5.1997 \times 10^{-4}$ |
| 1 | $-1.2134 \times 10^{-4}$ | $-2.5937 \times 10^{-5}$ | $-5.0341 \times 10^{-4}$ |
| 2 | $-1.1460 \times 10^{-4}$ | $-2.4630 \times 10^{-5}$ | $-4.7494 \times 10^{-4}$ |
| 5 | $-1.0136 \times 10^{-4}$ | $-2.1473 \times 10^{-5}$ | $-4.2142 \times 10^{-4}$ |
| 10 | $-9.1098 \times 10^{-5}$ | $-1.8893 \times 10^{-5}$ | $-3.8043 \times 10^{-4}$ |
| $\infty$ | $-4.3764 \times 10^{-5}$ | $-8.1624 \times 10^{-6}$ | $-1.8639 \times 10^{-4}$ |

From Table 26, one can observe that if the section is altered from the A to the B configuration, the evaluated displacement decreases by a factor between 1.5 and 1.86; if the section is altered to the C configuration, the displacement increases by a factor between 2.44 and 2.84. The rotation of the central node presented in Table 27 shows a decreasing trend by a factor between 4.6 and 5.4 when the section is altered from A to B. If section A is altered to C, the rotation increases a factor between 4.1 and 4.3.

## 4. Conclusions

From the results achieved, one can conclude that the base and the added material phases play an important role, particularly when the materials' dissimilarity is relevant, as is the case of the materials considered in the present study. In fact, if one considers the metallic phase as the base material and the ceramic as the added one, such inclusion results in improved structure performance. On the contrary, the mixture properties will worsen and degrade the structure's mechanical behaviour.

Overall, the nodal displacements show a decreasing trend with increasing ceramic phase quantity, which possesses a higher modulus of elasticity when compared to the metallic phase. Similarly, the natural frequencies present an opposite trend, although there is not a linear relation between frequency evolution and power law exponent progress. Specifically, in terms of the free vibration dynamic behaviour, if it is intended to maximize the fundamental frequency, it is visible that the best option for Truss 1, in terms of a functionally graded structure, is a power law exponent of 0.1, and the volume fraction is directly associated with the ceramic material. However, in this case, the all-ceramic structure enables a higher fundamental frequency. In the remaining cases, as symmetrical structures, it was noted that constructing the structures with homogeneous material is not the option that provides the best dynamic behaviour in the free vibration regime. For Truss 2, the best solution is a power law exponent of 0.1, but in this case, the volume fraction law describes the distribution of metallic material. For both Frame 1 and Frame 2, the best solution is a power law exponent of 2 and a volume fraction that describes the metallic material distribution.

The orientation of the cross-section significantly impacts the static and free vibration performance of the structures, which confirms our expectations.

Globally, the possibility of considering the variation in the materials' mixture ruled by structure-dependent volume fraction distributions can benefit the mechanical responses of structures and contribute to tailoring them toward specific operating conditions. These additional alternative or complementary solutions may enable improved free vibration and static performances, as shown by the case studies presented in this work.

**Author Contributions:** Conceptualisation, M.A.R.L. and J.I.B. Software, J.S.D.G. and M.A.R.L. Formal analysis, J.S.D.G., M.A.R.L. and J.I.B. Data curation, J.S.D.G. Writing—original draft preparation, J.S.D.G. Writing—review and editing, J.S.D.G., M.A.R.L. and J.I.B. Visualisation, J.S.D.G. Supervision, M.A.R.L. and J.I.B. All authors have read and agreed to the published version of the manuscript.

**Funding:** This research received no funding.

**Data Availability Statement:** The data generated or analysed are in the manuscript itself.

**Acknowledgments:** The authors acknowledge the support of FCT/MEC through Project IDMEC, LAETA UIDB/50022/2020. The authors also acknowledge the support of Project IPL/2022/VF_FGM_ISEL.

**Conflicts of Interest:** The authors declare no conflict of interest.

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
