# Peer review of "Static and Free Vibration Analyses of Functionally Graded Plane Structures"

_jcs, doi:10.3390/jcs7090377_

Round 1
Reviewer 1 Report
The article addresses an interesting topic, but the methods and basic idea is not current. The application of the solution to the chosen system is missing. Also, there is no justification why this type of plane system was solved. However, the biggest lack is the absence of a numerical approach of the FEM-based solution. Without this addition in the verification, this type of paper cannot be considered as scientific. Therefore, I recommend to revise the paper considerably and especially to supplement it with FEM models since experimental verification would obviously be a difficult task. When the numerical analysis is added, the number of references will increase as the number of references is low in the current form of the paper.
Author Response
Manuscript ID: jcs-2554389
Title: Static and Free Vibrations’ Analyses of Functionally Graded Plane Structures
Dear Reviewers
The authors are deeply grateful for the comments and suggestions received. The point to point response is attached

Reviewer 2 Report
The work can be accepted, but it is necessary to make appropriate corrections in order to get the work to the highest possible quality.
1. Functionally gradient materials are widely represented in contemporary scientific and professional literature. By changing the functionally guardant materials, especially composites, all the shortcomings of standard composites are eliminated. It is suggested that the authors expand the introduction with works in this area and explain the importance and need to consider these materials. Analyze at least 20 papers in the introduction part (see papers): http://hdl.handle.net/10204/6548 , https://doi.org/10.1016/j.compstruct.2021.113596, https://doi.org/10.3390/app10124190
2. Write please, what is the main contribution of this work and how this work differs from similar works in this field. Why should this work be published?
3. Labels in equation 1 and in other equations should be aligned in the text and in the equation. Especially indexes.
4. Why does a decimal point appear in Figures 1, 8, 11, 14 and 17 instead of a decimal point? Correct images in accordance with the instructions and standards. Unify the entire text.
5. Compare the results in tables 2 and 3 with the results of other authors, not with your own results. Why were these results taken?
6. Discuss the solutions obtained. What is their difference compared to the existing results? How applicable are these results? And for which materials have they proven to be the best?
7. Expand the concluding remarks based on the changes made, especially in the discussion.
Author Response

(The authors gave the same response as above.)

Round 2
Reviewer 1 Report
The paper was well improoved.
Reviewer 2 Report
the authors accepted all suggestions. I suggest paper for the press.